# Investigating whether deep learning models for co-folding learn the physics of protein-ligand interactions

Matthew R. Masters[1,2], Amr H. Mahmoud[1,2] & Markus A. Lill [1,2] ✉

Co-folding models represent a major innovation in deep-learning-based protein-ligand structure prediction. The recent publications of RoseTTAFold All-Atom, AlphaFold3, and others have shown high-quality results on predicting the structures of proteins interacting with small-molecules, nucleic-acids, and other proteins. Despite these advanced capabilities and broad potential, the current study presents critical findings that question the adherence of these models to fundamental physical principles. Through adversarial examples based on established physical, chemical, and biological principles, we demonstrate notable discrepancies in protein-ligand structural predictions when subjected to biologically and chemically plausible perturbations. These discrepancies reveal a significant divergence from expected physical behaviors, indicating potential overfitting to particular data features within its training corpus. Our findings underscore the models' limitations in generalizing effectively across diverse protein-ligand structures and highlight the necessity of integrating robust physical and chemical priors in the development of such predictive tools. The results advocate a measured reliance on deep-learning-based models for critical applications in drug discovery and protein engineering, where a deep understanding of the underlying physical and chemical properties is crucial.

The release of AlphaFold 2 (AF2), RoseTTAFold (RF), and related models marked a seminal moment in computational biology and revolutionized protein structure prediction. AF2 and RF were able to leverage evolutionary sequence and structural template data to create powerful deep learning models, capable of predicting proteins with breakthrough accuracy[1,2]. The impact of these models has been profound, spawning hundreds of subsequent papers including high-throughput structure prediction[3–5], ensemble sampling[6–8], molecular docking[9–12], stability prediction[13–16], further method development[17–20], and a host of other applied studies[21–24].

Recently, the works of AlphaFold 3 (AF3) and RoseTTAFold All-Atom (RFAA) have extended these capabilities to a broader array of biomolecular complexes, incorporating interactions with proteins, nucleic acids, and small molecules within a single predictive framework[25,26]. This approach, where the protein structure is predicted simultaneously with the ligand has been coined co-folding[27]. By using a diffusion-based architecture, AF3 was able to remove many of its complexities such as the stereochemical loss, amino-acid specific frames, and special handling of bonding patterns. The network also de-emphasises the importance of protein evolutionary data and opts for a more generalized, atomic interaction layer. These changes allowed AF3 to train on nearly all structural data which extended its modeling capabilities to new tasks, such as protein-ligand and protein-nucleic acid complexes.

When benchmarked against existing molecular docking tools, AF3 and RFAA showed markedly high performance. In terms of blind

[1]Department of Pharmaceutical Sciences, University of Basel, Basel, Switzerland. [2]Swiss Institute of Bioinformatics, Basel, Switzerland.
✉e-mail: markus.lill@unibas.ch

docking of small molecules to proteins with the PoseBusterV2 dataset[28], AF3 achieved an accuracy of around 81% for predicting the native pose within 2Å RMSD compared to the previous highest value obtained by DiffDock with 38%[29]. When the binding site is provided, traditional physics-based docking methods such as AutoDock Vina only reach an accuracy of about 60% compared to AF3 with over 93%[30]. This high accuracy is likely approaching experimental-level accuracy due to the dynamic nature of small molecules, especially in solvent exposed and variable regions[31]. These groundbreaking results highlight AF3's potential to disrupt conventional computational biology. However, they also raise questions about the physical robustness and generalization of these models in comparison to those based on physics.

Considering the accuracy of the results, these methods should adhere to the fundamental principles of physical interactions, such as hydrogen bonding, electrostatic forces, and steric constraints. These interactions govern molecular stability and specificity, making them essential for accurately predicting biologically relevant conformations. However, deep learning models primarily rely on data-driven pattern recognition, which does not necessarily equate to an understanding of physics. The ability to model physical interactions is crucial for real-world applications, particularly in drug discovery and protein engineering. The development of many small molecule medicines depend on precise atomic-scale modeling of protein-ligand binding, where small errors in structure prediction can lead to incorrect conclusions about biological activity, binding affinity, or specificity. If deep learning-based models fail to generalize beyond their training data and instead overfit to statistical correlations, they risk producing misleading predictions that do not translate into experimental success. Ensuring that these models respect physical constraints is therefore vital to their reliability and broader adoption in biomedical research.

In this paper, we aim to investigate the robustness of deep-learning-based co-folding models for predicting protein-ligand complexes by exposing the model to a series of adversarial examples. Our goal is to reveal the extent to which these models understand or fail to understand the underlying physical and chemical principles it is intended to simulate. By demonstrating the presence of vulnerabilities in these cutting-edge tools, we highlight the need for further improvements in the development of AI-driven structure prediction, ensuring that they are not only accurate but also generalizing to unseen protein-ligand systems and reliable in their modeling of molecular physics.

## Background and related work

The field of protein structure prediction has been revolutionized by deep learning models like AlphaFold 2 and RoseTTAFold, which significantly improved accuracy in predicting protein structures[1,2]. However, previous studies have shown that even small, biologically plausible perturbations can result in significant discrepancies in predicted structures, highlighting vulnerabilities in these models[32]. These findings resonate with broader machine learning research that shows it is common for deep learning models to robustly interpolate even in high-dimensional cases but fail to extrapolate when challenged by inputs not present during training[33,34].

The newly released versions of both AlphaFold and RoseTTAFold have moved towards diffusion-based approaches that aim to model arbitrary chemical structures under one unified framework and have shown impressive results[25,26]. Other open-source co-folding models, such as Chai-1[35] and Boltz-1[36], have optimized model architecture and performance, achieving AlphaFold3-level accuracy. These unified methods have been compared against several previous deep learning models which aimed to solve the task of protein-ligand docking directly, such as EquiBind[37] and DiffDock[29], as well as more conventional physics-based docking engines such as AutoDock Vina[30] and GOLD[38]. In addition to the targeted adversarial attacks demonstrated

against AF2, deep learning models for docking have also shown to be susceptible to non-physical artifacts, such as steric clashes and stretched bonds[29,39]. Furthermore, some studies have shown that the performance of these deep learning methods predominantly comes from their pocket finding ability and not an ability to resolve detailed molecular interactions[40,41]. Finally, deep learning models are also utilized for the scoring of docked poses, which have also come under scrutiny for their inherent bias and inability to understand physics[42,43].

The approach to generating adversarial attacks in prior studies relies on computationally searching for small perturbations to the input that produce a large change in the output. In order to find adversarial examples against AF2, researchers searched for mutations that lead to structures with high RMSD, while maintaining high BLOSUM similarity, helping to ensure that perturbations remain within biologically plausible limits[32]. This approach finds accuracy cliffs within the networks and demonstrates that they are not robust to their inputs. This flaw is present in nearly all deep neural networks without specific continuity constraints[44–46]. In terms of protein structure prediction, researchers found that predictions of fold-switched conformation are driven by structure memorization[47] and that AlphaFold is not learning the physics of protein folding[48]. In addition, other researchers have shown that co-folding models largely memorize ligands from their training data and do not generalize well to unseen ligand structures[49].

In this paper, we take a different approach to generating adversarial examples with a specific focus on the modeling of protein-ligand complexes. Rather than carefully crafted inputs that lead to a large output deviation, we crafted adversarial examples based on known physical, chemical, and biological first principles. For example, in the first challenge we selected all binding site residues forming contact with the ligand and mutated them to unrealistic substitutions that should displace the ligand from the binding site. In the subsequent challenges, we make modifications to the ligand which should also interrupt favorable interactions and drive the ligand out of the binding site. These examples have greater interpretability and are able to demonstrate other issues with the model, such as learning remote statistical correlations and its lack of physical understanding.

## Results
### Binding site mutagenesis

The results of the binding site mutagenesis challenges on ATP binding to Cyclin-dependent kinase 2 (CDK2) are shown in Fig. 1. In the results, wild-type always refers to the structure prediction made without any mutations from the original crystallized structure we are comparing against. For the unmutated wild-type, each of the four studied co-folding models successfully predicts the ATP binding mode, with AlphaFold3 performing the best (RMSD: 0.2Å) and RosettaFold All-Atom performing the worst (RMSD: 2.2Å). In the binding site removal challenge all binding site residues are replaced with glycine. All four co-folding models continue to predict the ATP-CDK2 complex with the same binding mode, despite the loss of all major side-chain interactions. There are no obvious unphysical clashes between the protein and ligand, however there are very few or no observed between the two entities. For example, the introduced mutations removed several of the positively charges residues that previously anchored the negatively charged ATP to the binding site. Although these residues have been removed, most of the co-folding models continue to place the ligand as if those favorable interactions are still present, indicated the models are overfit to this ATP-binding protein system. There are some minor changes: AlphaFold3 loses its precise placement, Boltz-1 places the triphosphate in a slightly different position, while Chai-1 remains mostly unchanged and RosettaFold All-Atom not necessary slightly improves accuracy (RMSD: 2.0Å).

The next challenge presents an even more dramatic change to the binding site, where all residues are mutated to phenylalanine,

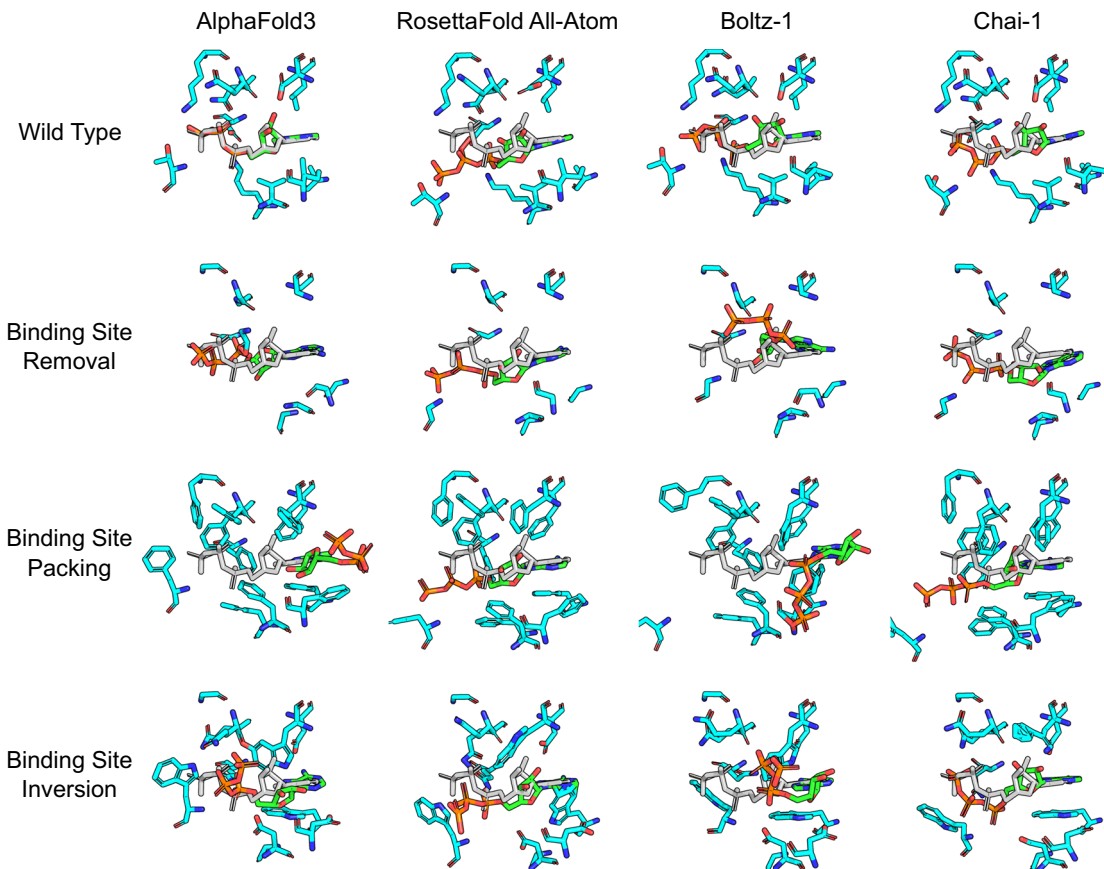

**Fig. 1 | Binding site mutagenesis challenges against co-folding models using the CDK2 system (PDB: 1B38).** Predicted binding-site residues are shown as cyan sticks, predicted ligand poses are shown as green sticks, and the original co-crystallized ligand pose is shown as gray sticks. The first row shows each model's prediction for the wild-type protein-ligand system prior to any modification. The remaining rows show different adversarial challenges where all binding site residues are mutated. In binding site removal, all residues are mutated to glycines effectively removing all ligand-side-chain interactions from the original system. The packing challenge mutates all residues to phenylalanine, removing all native interactions with side-chains and further occupying the pocket with bulky, hydrophobic groups. In the inversion challenge, binding site residues are mutated to residues with dissimilar properties. These mutations should annihilate the binding site and remove the majority of native protein-ligands interactions necessary for binding. However, in many cases the ligand is still predicted within the binding site and can adopt a low RMSD pose, indicating that these co-folding models are not predicting poses based on physics of interactions, but rather learning patterns in global protein structures and sequences.

effectively removing all favorable native interactions and occupying the space of the original binding pocket. The co-folding models demonstrate some capacity to adapt to these mutations, however the predictions are still heavily biased towards the original binding site, and in the case of RosettaFold All-Atom and Chai-1, the ATP molecule remains entirely within the binding site. There are also instances of unphysical overlapping atoms and large steric clashes in several of the predicted structures, indicating that under the time constraints of the diffusion process, the models are unable to either recognize or fully resolve the atomistic details. In an unbiased assessment of the protein-ligand structure, the binding site should be completely packed with the 11 phenylalanine rings, displacing the negatively charged ligand entirely from the pocket. However, even in the cases where the ligand pose is altered, it is still biased towards the ATP-binding pocket.

In the final binding site mutagenesis challenge each residue was mutated to a dissimilar residue, drastically altering the site's shape and chemical properties (see Section 'Binding Site Mutations' for details). The co-folding models lacked the ability to accurately respond to these mutations and continued to place the ATP molecule in the original binding site and do not significantly alter the binding pose. Once again, there were significant steric clashes between both protein and ligand atoms in several of the predictions, indicating that the full atomistic structure could not be fully resolved in time. Although Boltz-1 and

Chai-1 incorrectly placed the ligand within the original binding site, they also predicted a favorable stacking interaction between the adenine of the ligand and a tryptophan introduced during the mutation. Therefore these models still show a strong bias leading to unphysical predictions.

The CDK2-ATP complex was part of the training set used to develop the co-folding methods. To investigate whether the same observations applied to a system not included in the training data, we examined MEK1 bound to an inhibitor targeting the ATP binding site. The results of the binding site mutagenesis challenges on the MEK1 system are shown in Fig. 2. For the unmutated wild-type, three of the four co-folding models successfully predicts the ligand binding mode, with RFAA still placing near the binding site, but failing to retrieve a near-native pose. In the removal challenge, where all binding site residues are mutated to glycine, the results look similar with all but RFAA retrieving a near-native pose for the ligand. RFAA changes its prediction but is still inaccurate and contains numerous steric clashes between the ligand and surrounding residues. Additional attempts at rescuing RFAA accuracy by increasing the maximum number of cycles did not improve results. The trend continues in the binding site packing challenge where we again observe near-native poses without any significant deviation despite the dramatic changes to the pocket. Finally, the binding site inversion challenge

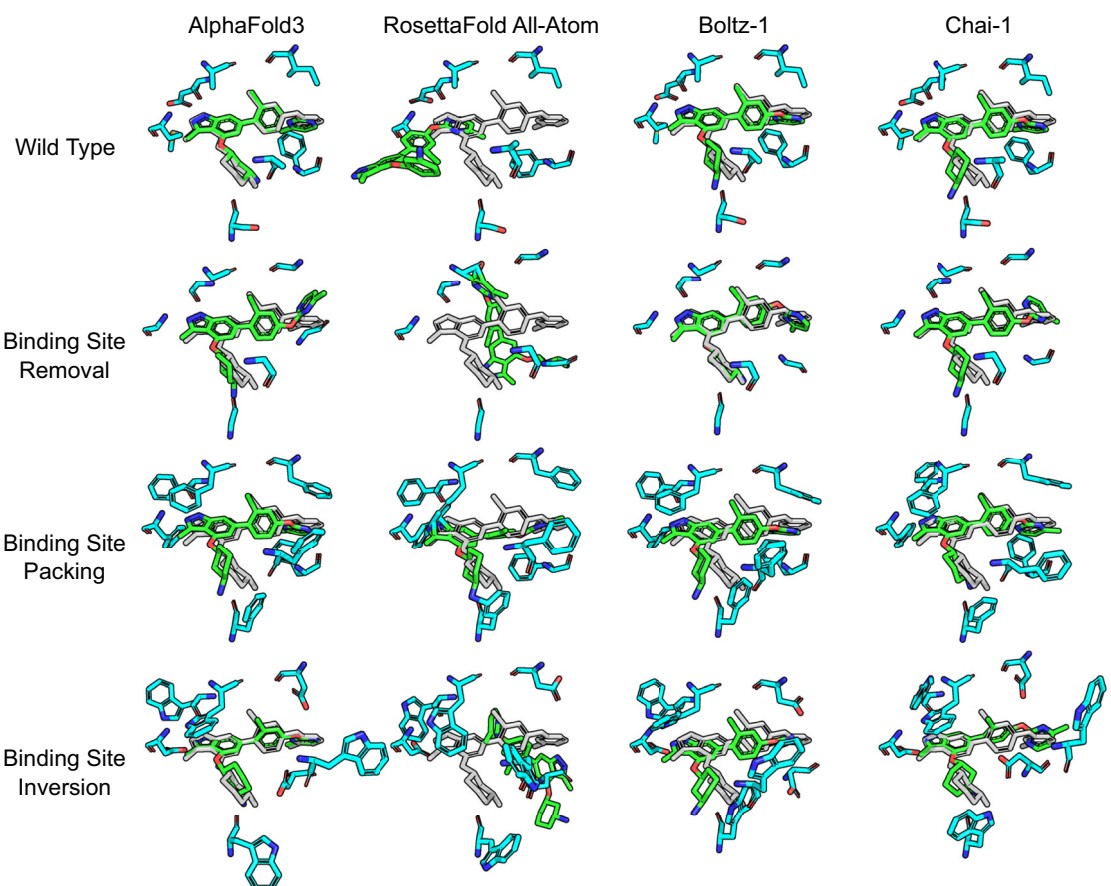

**Fig. 2 | Binding site mutagenesis challenges against co-folding models using the MEK1 system (PDB: 7XLP).** Predicted binding site residues are shown as cyan sticks, predicted ligand poses are shown as green sticks, and the original co-crystallized ligand pose is shown as gray sticks. Results are presented in the same manner as Fig. 1.

once again shows similar results with three of the co-folding models predicting almost the same pose each time, and RFAA failing to predict any good pose.

Overall, these challenges demonstrate that while co-folding models do capture some patterns from the data they are trained on, they are still heavily biased towards particular binding sites and do not ensure that the predicted structure are free of artifacts such as steric clashes or unfavorable interactions. Rather than learning about physical, atomistic interactions, the network seems to simply recognize patterns that are irrelevant for simulating the physical system, such as specific ordering of the sequence. With the exception of binding-site sidechains, the tertiary protein structure was also predicted with consistently high accuracy, further suggesting that these predictions are not sensitive to adversarial changes in the protein constitution. Additional 2D and 3D visualizations of the protein-ligand interactions of the mutated structures are provided in Supplementary Information Figs. S1 and S2.

In addition to the predicted structure produced by each of the co-folding models, each structure is also evaluated using a confidence module trained in conjunction with the structural diffusion model. The confidence metrics, including ligand pLDDT, ligand pTM, and protein-ligand ipTM, are reported in the Supplementary Information Tables S1 and S2 for the CDK2 and MEK1 systems, respectively. The local distance difference test (LDDT) is a metric for evaluating local quality via distance differences in a fully atomistic model, even when conformational changes are present[50]. Protein structure prediction models have been trained to predict the calculated LDDT, thus giving a per-residue metric for how confident the model, termed predicted LDDT (pLDDT)[1]. The metric has shown to be useful for distinguishing

high quality from low quality structures and more structured regions from more disordered regions[51]. Generally, a pLDDT above 70 is considered high confidence, indicating that the backbone is predicted correctly, although side chains may be somewhat misplaced. A pLDDT above 90 is regarded as highly accurate and very likely to agree closely with the experimental structure. All of the investigated co-folding models extend this metric to include ligands, predicting a pLDDT value for each ligand atom (per token). While it remains to be determined exactly how to interpret these values in the context of individual atoms rather than protein residues, the same general trend appears to hold: scores above 70 indicate high confidence, and scores above 90 reflect very high confidence.

In order to further confirm this behavior as a widespread issue for protein-ligand structure predictions, the binding site mutation methodology has been applied systematically to the CASF-2016 dataset. In each system, binding site residues were automatically recognized based on a distance criterion and were mutated based on the three challenges (removal, packing, and inversion), and subsequently predicted using AlphaFold3. The results of this study are presented in Fig. 3. Overall, the trend holds and the same behavior is observed across most systems present in the dataset. Irrespective of the type of modification, more than half of all systems correctly predicted by AlphaFold3 (ligand RMSD < 2 Å) retain the same predicted ligand pose even after binding site disruption. This proportion of conserved binding modes is largely independent of the confidence score. For example, among complexes with a confidence score > 80, between 42% and 52% maintain an unaltered ligand pose depending on the disruption applied (compared to 78% unaltered in the absence of disruption).

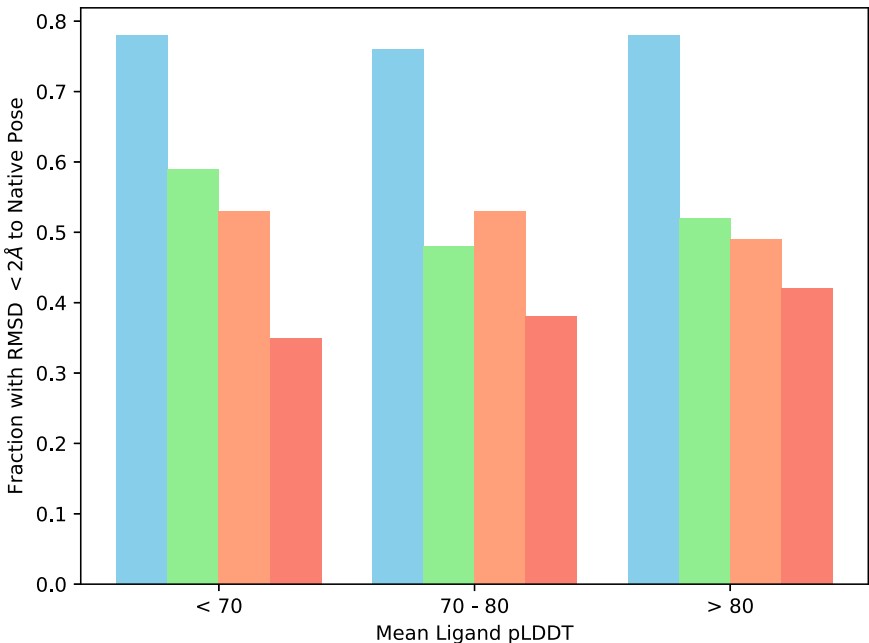

**Fig. 3 | Systematic binding site mutation challenges performed across the CASF-2016 protein-ligand dataset (*n*=285 complexes).** Bar represents fraction of systems with RMSD < 2 Å to the native pose for the unmutated wild-type (blue), binding site removal (green), binding site packing (orange), and binding site inversion (red) challenges.

## Ligand modifications

In addition to the binding site mutagenesis challenges, we designed two ligand-focused challenges to assess how the co-folding models handle modifications to the ligand itself. In the first challenge, a sugar ligand was progressively methylated, reducing its hydrogen bonding capacity and introducing additional steric hindrance. Similar to the previous challenges, we tested the progressively methylated sugar ligand with all four co-folding models with results shown in Fig. 4. For the unmutated wild-type, each of the co-folding models successfully predicts the binding site of the sugar molecule. However, while AlphaFold3 and Chai-1 were able to predict the precise orientation of the sugar ring, RFAA and Boltz-1 predicted a rotated binding mode in which the sugar ring was shifted to different ring positions (i.e., the first carbon was misaligned to the second and fourth positions, respectively). No obvious steric clashes or non-physical artifacts were observed in the wild-type predicted structures. Following the first methyl addition, AlphaFold3, RFAA, and Boltz-1 showed minimal changes in the predicted binding mode, whereas Chai-1 displaced the ligand completely, suggesting a heightened sensitivity to this modification. This indicates that Chai-1 is particularly sensitive to this modification and is able to respond in the expected way immediately. With each additional methyl group, AlphaFold3 and RFAA gradually moved the ligand out of its original pocket, although it remained nearby, suggesting the model still has a preference for this region. By contrast, Boltz-1 was the slowest to respond to these mutations and still places the fully methylated ligand close to the original site, whereas Chai-1 consistently displaced the sugar throughout the entire methylation series. Several steric clashes and stretched bonds were observed between the ligand, co-factor, and protein in the RFAA predictions with the addition of methyls. All other predictions remained free of any obvious clashes or unrealistic molecular geometry.

We next examined the co-folding models' ability to accommodate major charge variations in ligands by modifying ATP with a net charge of −3 to related molecules with charge states ranging from 0 to +3. These changes were performed by replacing the negatively-charged triphosphate by either alkyl chains with increasing length of positively-charged quaternary amine groups, shown in Fig. 5B and C,

respectively. Such drastic changes should cause significant changes to the binding mode due to the presence of several positively charged groups that cause electrostatic repulsion with the binding site. Surprisingly, converting the triphosphate into a positively-charged group had little effect on the predicted binding modes across all four models. As further quaternary amine groups are added, the ligand becomes more positively charged and should be repelled from the positively charged ATP binding pocket. By the final addition, AlphaFold3 and Boltz-1 both partially displaced the ligand from the binding site, showing some understanding of electrostatic interactions. The ligands are not fully displaced and the adenine ring is still bound, showing that the ligand still has a preference for this pocket, despite other favorable interactions likely present elsewhere on the protein. By contrast, RosettaFold All-Atom and Chai-1 showed little to no repositioning in response to the extreme shifts in charge, indicating they may be less reliable for modeling electrostatic interactions and remain biased toward the ATP binding pocket. No steric clashes or unrealistic molecular geometry was observed in any of the charge-modification predictions. Additional 2D and 3D visualizations of the protein-ligand interactions can be found in Supplementary Information Figs. S3 and S4 for the sugar and charge modifications, respectively. Furthermore, the confidence metrics for these two systems are reported in the Supplementary Information Tables S3 and S4. Tanimoto similarity coefficients were calculated for each modified ligand structure compared to the unmodified original and are presented in Supplementary Information Table S5.

It should be noted that in contrast to classical docking methods where the binding pose is typically limited to a predefined search volume, co-folding methods are in principle able to place the ligand anywhere on the protein surface. However, each of the tested co-folding models demonstrated a specific bias towards a given pocket rather than exploring other, potentially unspecific binding modes on the surface.

To proof the existence of the discussed bias of co-folding methods for placing the ligand to the original binding site despite drastic alterations of the pocket, we performed extensive funnel-metadynamics studies for each of the mutated cases presented here.

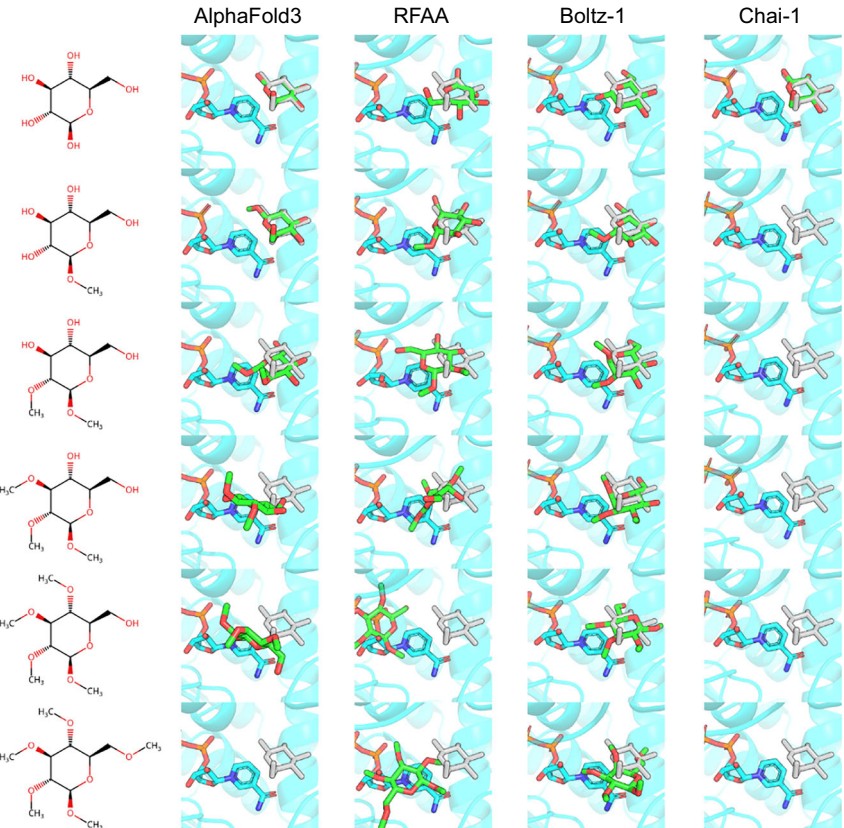

**Fig. 4 | Ligand methylation challenge against co-folding models using the Glucose Dehydrogenase system (PDB: 2VWH).** The first row shows predictions of the native structure with no changes to the ligand. Subsequent rows add a methyl group to the glucose alcohols, removing hydrogen bond donors and its ability to interact favorably with its binding pocket. Predicted structure shown in cyan (protein and cofactor) and green (ligand). Reference ligand position from crystal structure shown in gray.

Funnel metadynamics is a variant of enhanced sampling wherein a restraining funnel-shaped potential is introduced around the binding site. This setup focuses sampling on the transition between bound and unbound states while reducing extraneous exploration of the solvent-exposed protein surface. By monitoring the free energy as the ligand moves in or out of the binding pocket, we can determine whether it is thermodynamically favorable for the ligand to remain bound under the imposed mutations or modifications. In other words, if the protein-ligand interaction is physically unrealistic, for example, due to large steric clashes or impossible electrostatic arrangements, the funnel-metadynamics simulations will favor the unbound state, generating a low probability for stable binding. Consequently, measuring these probabilities allows a direct comparison with the co-folding predictions, confirming whether the model's structural output is supported by underlying energetics or not.

Table 1 displays the predicted probability differences between binding of the ligands to the various binding sites. This highlights that in the case of binding site packing and inversion no binding can be expected in contrast to the co-folding predictions. The results agree with the expectation that the mutations greatly diminish binding, and in many cases, prevent it entirely. In the case of the binding site removal challenge, we continue to see some binding, which is lost completely when the residues are mutated further in the packing and inversion challenges. This is explained by the fact that the ligand maintains several backbone contacts and has little steric hindrance to entering the site. In the packing and inversion challenges, the binding site becomes so crowded and disrupted as to prevent all ligand binding. In the glucose challenges, we observe a steep drop off in the binding with the addition of methyl groups, agreeing with our expectation. In the ATP charge modification examples, we do not observe any reproduction of the original crystal pose at all. These results demonstrate that ligand placement by co-folding methods in significantly altered binding pockets is often physically incorrect. This highlights the tendency of these methods to position ligands based on overall protein sequence rather than physicochemical principles. Plots of the ligand RMSD time evolution throughout each of the simulations is shown in Supplementary Information Figs. S5 and S6.

## Discussion

Our study provides insight into into the performance of co-folding models by combining data-driven predictions with physically motivated adversarial challenges. Traditional benchmarks primarily measure how well these models capture structural patterns present in training datasets, but our work exposes the extent to which they actually capture underlying biochemical and biophysical constraints. In doing so, we bridge a critical gap in the current literature: while co-folding methods have demonstrated high accuracy for tasks such as blind docking or binding site identification, there has been limited exploration of whether they truly learn the physics of molecular recognition. By systematically mutating binding site residues or ligand functional groups, we reveal that the models are prone to placing ligands in pockets despite clearly unfavorable electrostatic or steric interactions.

Our results demonstrate that current co-folding models, despite their impressive performance under normal conditions, exhibit notable limitations when confronted with intentional perturbations—both on the protein binding site and the ligands themselves. Across our binding site and ligand modification challenges, we observed a

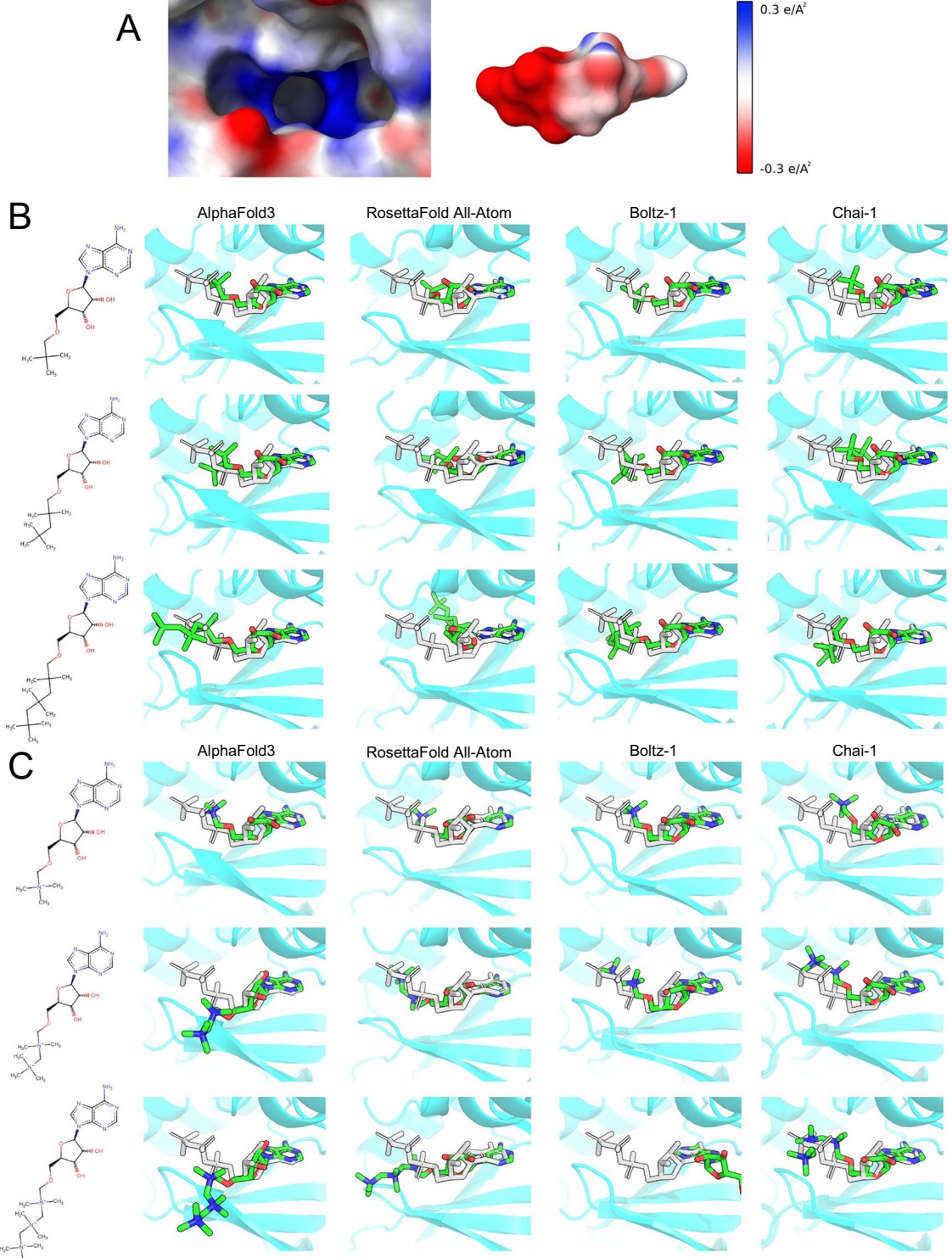

**Fig. 5 | Modifying charge challenge against co-folding models using the CDK2 system (PDB: 1B38). A** Electrostatic surface of the crystal structure pocket and unmodified ATP molecules, demonstrating the complementarity positive and negatively charged surfaces. **B** Co-folding predictions where phosphate groups are replaced by uncharged tert-butyl groups. **C** Co-folding predictions where phosphate groups are replaced by positively charged quaternary amine groups. The first row shows predictions of adenine choline (+1 charge) in protein (cyan) and ligand (green) with the reference ATP ligand (gray). Subsequent rows add an additional choline group, further increasing the formal charge, which should act as a strong repulsive force to displace the ligand (or at least the choline) from the triphosphate binding site.

consistent bias toward perserving the original binding geometry, even when significant structural, chemical, and physical changes were introduced. This highlights only a partial capacity to adapt to disruptions, evidenced by instances of partial ligand displacement or the formation of new interactions, but also underscores the models' struggles to fully capture realistic atomistic details when faced with unique prediction tasks. The failure of these models is unsurprising when considering the featurization they rely on in order to make their predictions. Most of the co-folding models rely heavily on multiple-sequence alignment and 3D template-based input features retrieved

**Table 1 | Funnel metadynamics results for each of the tested systems showing probability of ligand being found in the bound or unbound state**

| System | Bound State | Unbound State |
|---|---|---|
| **CDK2: Binding site modifications** | | |
| Wild-Type | 0.61 ± 0.05 | 0.39 ± 0.05 |
| Binding Site Removed | 0.42 ± 0.04 | 0.58 ± 0.04 |
| Binding Site Packed | 0.00 ± 0.00 | 1.00 ± 0.00 |
| Binding Site Inversion | 0.00 ± 0.00 | 1.00 ± 0.00 |
| **Glucose Dehydrogenase: Ligand methylation** | | |
| Glucose Unmodified | 0.91 ± 0.06 | 0.09 ± 0.06 |
| Glucose 1 Methyls | 0.81 ± 0.06 | 0.19 ± 0.06 |
| Glucose 2 Methyls | 0.06 ± 0.04 | 0.94 ± 0.04 |
| Glucose 3 Methyls | 0.00 ± 0.00 | 1.00 ± 0.00 |
| Glucose 4 Methyls | 0.00 ± 0.00 | 1.00 ± 0.00 |
| Glucose 5 Methyls | 0.00 ± 0.00 | 1.00 ± 0.00 |
| Glucose 6 Methyls | 0.00 ± 0.00 | 1.00 ± 0.00 |
| **CDK2: Phosphate replacements on ATP** | | |
| ATP (Charge +1) | 0.00 ± 0.00 | 1.00 ± 0.00 |
| ATP (Charge +2) | 0.00 ± 0.00 | 1.00 ± 0.00 |
| ATP (Charge +3) | 0.00 ± 0.00 | 1.00 ± 0.00 |

Simulations were run in triplicate and data is presented as mean ± standard deviation.

from existing data which help the model predict the 3D structure. However, when we introduce small mutations, for example by removing the binding site, it is very likely that the sequence alignment and template search will return exactly the same results as before, as they are still the closest related sequences and structures in the dataset. Therefore, the MSA and template features that the network accepts as input are identical despite the mutations, leading the model to produce a very similar prediction.

In addition to the results presented in the main article, additional analysis was performed in order to investigate what, if any, factors influence the models' likelihood to consistently place the ligand in the same binding mode. We investigated the per-system performance across the three challenges compared to wild type and predicted confidence (ligand pLDDT). These results are shown in the Supplementary Information Figs. S7–S12. In addition, we investigated whether belonging to particular protein families played a significant role by grouping results from the CASF predictions into their protein families. These results, presented in Supplementary Information Figs. S13 and S14, show no clear separation between protein families that perform robustly or not. Additionally, no correlation between the number of examples in the training set versus model sensitivity against adversarial changes was observed. In another analysis, shown in Supplementary Information Fig. S15, we investigated whether certain physicochemical properties had an influence on the robustness of the results, such as pocket solvent-accessible surface area, ligand molecular weight, polarity, and solubility. However, the results were similar and there was no clear method to distinguish systems which are susceptible to this failure mode or not. Finally, we performed additional predictions of the mutated CASF proteins in the absence of a ligand to assess its influence on the protein structure. The results, shown in Supplementary Information Fig. S16, demonstrate that the ligand has little influence on the overall fold of the protein, with 95% of CASF structures having an apo structure with $C_\alpha$ RMSD less than 1.5 Å to the holo structure.

These findings have important implications for both users and developers of co-folding models. We advocate for a cautious reliance on deep-learning-based models for critical applications in drug discovery and protein engineering, where a deep understanding of the

underlying physical and chemical properties is crucial. Users should be aware that the accuracy of these models may degrade substantially when predicting non-standard or previously unseen protein-ligand systems, and that additional validation such as physics-based methods, such as molecular dynamics simulations or free energy calculations, or experimental data may be necessary. Beyond the specific adversarial tests employed here, a comprehensive validation pipeline could involve multi-tiered assessments. First, simple heuristic checks such as counting the number of steric clashes or evaluating the electrostatic potential can serve as rapid proxies for physical realism. Second, repeated predictions under randomized seeds or minor input perturbations can reveal how sensitive each model is to local minima in the folding landscape. Finally, high-level metrics akin to those used in protein design (e.g., Rosetta energy scores or MD-based free energy calculations) could offer more rigorous confirmations. Such a multi-faceted approach would not only highlight where co-folding tools excel but also provide invaluable insight into their failure modes.

For developers of these tools, our results point to the need for better physics integration and further methodological improvements in order to generalize accurately across chemical and sequence space. In particular, future co-folding models should obey known physical principles such as van der Waals repulsion and Columbic electrostatics, thereby producing structures free of steric clashes or unresolved high-energy interactions. Future co-folding models should assess not only their performance on common benchmarks, but also how the model responds to modifications to the protein or ligand and observe if the predictions agree with expert intuition. Nevertheless, further systematic evaluations, improvements to the models' methodology, and strong integration of physics will be crucial for enabling future co-folding models to reliably handle any type of biomolecular prediction and drive continued progress in this rapidly evolving field.

Our results demonstrate that current co-folding models, while highly accurate across systems contained in the training data, do not function as robust physics-based systems; they can generate non-physical, high-energy structures with considerable confidence. By attempting to displace ligands through the removal of native contacts or the introduction of unfavorable interactions, we observed that all four examined models frequently maintained nearly identical ligand positions, despite lacking meaningful interactions to support them. Nevertheless, these tools represent an impressive paradigm shift in biomolecular modeling at scale, offering rapid structural predictions that, in many cases, are useful starting points for hypothesis generation in biology, chemistry, and medicine.

However, we recommend that researchers treat these predictions with caution and supplement them with additional, physics-based analyses—such as molecular dynamics simulations or more rigorous free energy calculations—before relying on them for critical applications. In the same vein, future adversarial challenges should encompass solvent effects, multiple ligand conformational states, and chemically diverse scaffolds to further test and refine model capabilities.

Ultimately, while co-folding models show groundbreaking potential for predicting complex biomolecular assemblies, our study highlights their inconsistencies with fundamental chemical and physical principles. Such vulnerabilities underscore the importance of validating model outputs against established scientific principles, particularly in high-stakes areas like drug design. Addressing biases introduced by training data and improving generalization to novel chemical entities will be pivotal steps toward building truly reliable and broadly applicable co-folding models.

## Methods

In this study, we perform several types of adversarial challenges by crafting examples with an expected physical outcome and seeing if the predictions from co-folding models agree with this expectation. To

this end, we employed four state-of-the-art co-folding models: AlphaFold3[25], RosettaFold All-Atom[26], Chai-1[35], and Boltz-1[36]. These models each demonstrates impressive and competitive results on a variety of benchmarks. However, the physical robustness of these models to adversarial challenges has yet to be tested or compared. Unless otherwise noted, each models was run using default settings and no constraints were specified to the models that support this feature.

## Binding site mutations

**Removing interacting residues.** In this challenge, we identify residue side-chains that interact with a ligand and mutate them into glycine residues, effectively removing the driving force for binding and ligand recognition. For this challenge, the ATP-binding protein CDK2 (PDB: 1B38) and MEK1 (PDB: 7XLP) were selected as the test systems. ATP is a good test case since it is widely represented in the PDB training data and its binding involves strong enthalpic interactions due to its negatively charged phosphate groups, anchored by forming salt-bridges with multiple positively-charged side-chains. MEK1 was selected as an additional test case since it was not included in the training set of any of the tested co-folding models and has low similarity in terms of ligand and pocket shape. The system was identified via the Runs N' Poses dataset[49] which calculated a similarity value (`sucos_shape_pocket_qcov`) of 46.21. Residues with side-chain heavy atoms within 3.5 Å of any ligand heavy atoms were selected for mutation. In the case of CDK2, these residues were: I10, T14, V18, A31, K33, D86, K129, Q131, N132, L134, D145. In the case of MEK1, these residues were: A40, A59, I105, E108, M110, S158, and F173. Following mutagenesis of the sequence, structures were generated with their respective ligands using each of the co-folding tools.

**Packing with bulky hydrophobes.** In addition to the previous approach where we replace critical residues with glycine, we also mutated the same residues with phenylalanine residues. This task not only destroys the interactions that drive ligand binding, but also occupies the binding site with bulky, hydrophobic groups. Based on physico-chemical knowledge, we would expect these groups to avoid contact with the solvent, and especially highly polar groups like the triphosphate of the ATP. The aim of this challenge is to mimic the hydrophobic effect and provide an additional penalty to try and displace the ligands from their binding pockets. We used the same test system and mutation locations as the previous challenge.

**Mutating to dissimilar residues.** In a final binding site mutagenesis challenge, we chose to mutate residues into those with opposing properties. For example, a residue with a small, polar side-chain (e.g., serine) is replaced by a large, non-polar residue (e.g., tryptophan), and vice-versa. This challenge goes one step further, not only removing all interactions or crowding the binding site, but actually replacing favorable interactions with unfavorable ones. To this end, we utilized the Miyata distance which gives a quantitative measure of how similar two amino acids are based on two properties: volume and polarity[52].

As described in the main text, one of the binding site mutagenesis challenges we employed was to replace contacting residues with those that have opposite properties. In this study, we utilized the Miyata distance in order to select mutations with significant dissimilarity. The Miyata distance is defined as

$$d_{ij} = \sqrt{\left(\frac{\Delta p_{ij}}{\sigma_p}\right)^2 + \left(\frac{\Delta v_{ij}}{\sigma_v}\right)^2} \quad (1)$$

where $\Delta p_{ij}$ is the difference in polarity between replaced amino acids, $\Delta v_{ij}$ is the difference in volume, and $\sigma_p$ and $\sigma_v$ are the standard

**Table 2 | Assignment for mutating to dissimilar binding site residues according to the Miyata distance**

| Wild-Type | Mutation | Distance |
|---|---|---|
| TYR | GLY | 4.08 |
| TRP | GLY | 5.13 |
| LYS | GLY | 3.54 |
| ARG | GLY | 3.58 |
| VAL | ASP | 3.40 |
| LEU | ASP | 4.10 |
| ILE | ASP | 3.98 |
| MET | ASP | 3.69 |
| PHE | ASP | 4.27 |
| ALA | TRP | 4.23 |
| GLY | TRP | 5.13 |
| PRO | TRP | 4.17 |
| HIS | TRP | 3.16 |
| ASP | TRP | 4.88 |
| GLU | TRP | 4.08 |
| CYS | TRP | 3.34 |
| ASN | TRP | 4.39 |
| GLN | TRP | 3.42 |
| THR | TRP | 3.50 |
| SER | TRP | 4.38 |

deviations for $\Delta p_{ij}$ and $\Delta v_{ij}$, respectively. Therefore, we summarize the mutations we applied during this challenge in Table 2.

Therefore, for the ATP-binding protein Human Cyclin-Dependent Kinase 2 (CDK2), the introduced mutations were: I10D, T14W, V18D, A31W, K33G, D86W, K129G, Q131W, N132W, L134D, and D145W. For the MEK1 system, these mutations were: A40W, A59W, I105D, E108W, M110D, S158W, and F173D. All structural predictions for the binding site mutation challenges on CDK2 and MEK1 are shown in Figs. 1 and 2. Three additional binding site mutagenesis challenges performed on AF3 are shown in the Supplementary Information Fig. S17.

**CASF-2016 dataset.** In addition to the CDK2 and MEK1 case studies, we extended our investigation to perform these binding site mutation challenges against a dataset of high-quality, experimentally-determined protein-ligand complexes. The CASF-2016 dataset contains 285 protein-ligand complexes and is a common test set for docking and structure prediction methods[53]. Binding site residues were identified with the co-crystallized ligand using the same distance threshold of 3.5 Å. Structures were predicted using only the Alpha-Fold3 co-folding model and the highest-confidence structure was used for analysis.

## Ligand mutations

**Methylation.** In addition to the mutations done from the perspective of the protein sequence, we also experimented with mutations to the ligand input. The first of these challenges is methylation, by which we add methyl groups onto polar sites of the ligand, removing its ability to form hydrogen bonds and adding steric hindrance to its binding. For this example, we utilize the Glucose Dehydrogenase system (PDB: 2VWH). Glucose is a good test case since sugar structures are widely represented in the PDB training data and its binding is dominated by its hydrogen bonding due to the presence of five alcohol groups. In this test, we progressively methylate each of the alcohols beginning from the first position and predict structures for each protein-ligand complex. Methylation of glucose can be assumed to result in non-binders that lose all affinity to the target due to its loss of ability to donate

hydrogen bonds and the steric hindrance of the additional methyl groups. The predictions from each tool are shown in Fig. 4.

**Modifying charge.** In addition to the methylation challenge which disrupts favorable electrostatic interactions, we can also replace charged groups of a ligand to test how cofolding models respond to changes in the formal charge of the ligand. For this challenge, we again adopted the CDK2 system, since it binds the negatively-charged ATP molecule which we can transform into a positive charge. To this end, the triphosphate group was replaced with different numbers of tert-butyl and positively-charged quaternary amine groups, respectively. This allows the construction of test molecules with formal charges 0, +1, +2, and +3 that share a structural resemblance and topological equivalence to the negatively-charged adenosine phosphates. As with previous challenges, the assumption is that such a strong difference in charge should show a significantly different binding mode of at least the charged groups, if not complete loss of binding. The predictions for this challenge are shown in Fig. 5.

### Funnel metadynamics simulations

Funnel metadynamics simulations were utilized in order to further validate the unphysical nature of the co-folding predictions of the modified protein-ligand systems. To this end, funnel metadynamics simulations were set up so that the ligand is initially placed at the start of a funnel encompassing the full binding site. When applied to a protein-ligand system with accessible binding site and high affinity, the ligand should show binding and unbinding events during the simulation, allowing for an accurate estimate of the free energy of binding and differences in the relative densities of the states. Funnel metadynamics simulations were setup using the Biosimspace package, relying on OpenMM for the underlying simulations[54,55]. The Amber ff14SB force field was used to parameterize the protein, TIP3P was used for the water model, and GAFF2 was used to parameterize the ligand[56–58]. The system was solvated in a water box with padding of 15 Å. For each protein target, the funnel parameters were kept the same and ensured that the binding site is accessible to the solvent and is not buried. The simulations were run for a total of 1000 ns each in triplicate to ensure robustness of the results. Correction for the loss of rotational and translational freedom of the ligand due to the funnel restraints are calculated using the methodology of Rhys et al.[59]. Convergence was observed by plotting the density as a function of the projection along the collective variable. The bound state was defined by having RMSD < 2 Å to the native pose, all other stable states outside of the pocket are considered to be unbound. In the modified ligand challenges, the RMSD calculation was performed across corresponding heavy atoms. For example, in the methylation challenge, only the original heavy atoms of the sugar contribute to the RMSD, not the additional methyl groups. In the charge modification example, the charged groups were included in the RMSD calculation.

### Reporting summary

Further information on research design is available in the Nature Portfolio Reporting Summary linked to this article.

## Data availability

All predicted structure files are available on Zenodo at (https://zenodo.org/records/14749304). Note that some of the output data is licensed under strict terms-of-use and has several restrictions that should be respected.

## Code availability

Code associated with running the co-folding models is available at (https://zenodo.org/records/14768184) and is distributed under an MIT license.

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

## Acknowledgements

The work was partially supported by the Swiss National Science Foundation (Project number: 310030_197629) and the Novartis Research Foundation.

## Author contributions

M.R.M.: conceptualization, methodology, investigation, formal analysis, writing & review. A.H.M.: conceptualization, interpretation, writing & review. M.A.L.: conceptualization, interpretation, writing & review. All authors approved the final manuscript.

## Competing interests

The authors declare no competing interests.
