## [Transparent Peer Review file · Nature Communications]

Investigating Whether Deep Learning Models for Co-Folding Learn the Physics of Protein-Ligand Interactions

Corresponding Author: Professor Markus Lil

Version 0:

Reviewer comments:

Reviewer #1

(Remarks to the Author)

In this manuscript, the authors investigate the limitations of AF3 and three other AF3-like methods in predicting protein-ligand interaction structures. By introducing in silico modifications designed to disrupt known interactions, they demonstrate that these methods often create unphysical poses mimicking corresponding training poses of the original ligands, highlighting potential pitfalls in protein-ligand interaction predictions.

Since AF3 and related methods were designed to predict structure for protein-ligand pairs that are known to interact, their inability to distinguish true interactors from artificially designed non-interactors (as tested here) is perhaps unsurprising. As the authors note, a key limitation is the scarcity of training data, a well-recognized challenge in the field. However, for practical applications, the confidence metrics provided by these methods are crucial and should be considered in the benchmark assessment. Low-confidence predictions could alert users to unreliable results, preventing misinterpretation. While the authors include these metrics in supplementary tables, they do not mention nor discuss them at all. A deeper analysis of confidence scores versus the degree of adverse modification would strengthen the study.

A complicating factor to this study is the biological robustness of protein-ligand systems: some modifications may be physically and functionally tolerated, while others may not. The authors use funnel-metadynamics simulations to assess bound states of their modified ligands, and some of the modifications still permit (or even favor) bound states according to their simulations. In such cases, are the AF3 predictions physically plausible? This distinction warrants further discussion.

In the Discussion, the authors propose that Rosetta energy scores or MD free energy calculations could provide "more rigorous confirmation." While cross-validation with physics-based methods may be useful, these approaches have well-known limitations. Otherwise, deep learning would not lead the field now. A more pertinent question is whether these methods can complement each other, which this study does not explore sufficiently. While the adversarial cases here can be dismissed due to obvious clashes, what about physically plausible predictions without any clashes for novel ligands? Are they hallucinations or realistic predictions?

Minor Comments:

1. Please include at least one reference for funnel-metadynamics.
2. In Figures 1–3, consider annotating unphysical interactions/clashes for clarity.
3. In Figure S1, the labels are too small to read. Please also expand the caption to explain the color scheme and mark any unphysical interactions, if present.

Reviewer #2

(Remarks to the Author)

The manuscript by Masters, Mahmoud and Lill addresses one of the most important problems in modern AI for biomolecular modelling: whether the latest advances in co-folding (as in RoseTTAFold-AllAtom, AlphaFold 3, and others) capture valuable physics of protein-ligand interactions, thus being able to generalise to complex cases, or whether they are mostly memorising and overfitting. The authors test this hypothesis by generating "adversarial" examples where the model betrays having memorised examples from the data, such as removing all interactable hotspots from a pocket, or filling it with bulky

amino acids; as well as lightly modifying ligands to remove functional groups that take part on crucial interactions. The authors explore several adversarial examples on two systems, CDK2 and glucose dehydrogenase, and use them to conclude that modern co-folding approaches are not correctly capturing the physics of protein-ligand interactions.

The authors present an elegant and insightful methodology to test the models, and provide a valuable contribution to the field. In my opinion, the article is technically sound and well-written, and I believe that it should eventually be published in Nature Communications, however I believe that there are some points (two minor and one major) that need to be addressed before we can reach a final decision of acceptance.

The first, major point is that the article is at best a compilation of two case studies. While I find the authors' methodology and results very compelling, I would be hard-pressed to present this article to a colleague to support the conclusions -- there simply isn't enough evidence. The conclusions of the article would be far stronger if they repeated some of their experiments on multiple proteins and ligands. Even if they repeated the binding site removed/packed experiments on all crystal structures on CASF16 (~200 structures), that would be enough to provide a broad view across target classes and compounds (though I would probably suggest that the authors also explore more ligand modifications). This experiment would be trivial to implement and require at most a couple of GPU days with even a gaming-grade GPU, so I cannot see any reason to not perform it. In my opinion, as much as I think this is a promising article with a valuable message for the molecular biology community, without a broader set of experiments I could not in good faith recommend publication on Nature Communications, or other peer-reviewed journals.

The second and third point are more minor. First, the authors have not fully disclosed the methodology: there is no mention to how the funnel metadynamics simulations were conducted, which is important to the conclusions of the article. Second, the authors have forgotten to cite some important literature in the area of studying the physical underpinnings of AI structure prediction (some of which include my own work -- sorry, I hate to be that guy):

Outeiral, C., Nissley, D.A. and Deane, C.M., 2022. Current structure predictors are not learning the physics of protein folding. *Bioinformatics*, 38(7), pp.1881-1887.

Chakravarty, D., Schafer, J.W., Chen, E.A., Thole, J.F., Ronish, L.A., Lee, M. and Porter, L.L., 2024. AlphaFold predictions of fold-switched conformations are driven by structure memorization. *Nature communications*, 15(1), p.7296.

I look forward to read the authors' improved article.

Carlos Outeiral

Reviewer #3

(Remarks to the Author)

In this manuscript, the authors claim that deep learning models such as AlphaFold3 (AF3) fail to learn the physics of protein-ligand interactions and predict structures with a strong bias toward the training data. However, there are a number of concerns with this manuscript, as described below.

1. In the AF prediction, the main chain structure is expected to be correct when pLDDT > 70, and both the main chain and side chain structures are expected to be correct when pLDDT > 90. Supplementary Tables S2-S4 show that in the present predictions, pLDDT was greater than 90 for the wild type. In contrast, pLDDT was less than 90 for the structures predicted when amino acid substitutions or ligand modifications are introduced, suggesting that side-chain and ligand structures are unreliable for these predictions. The authors made the argument that "physics does not hold" with unreliable ligand structures. However, the argument about unreliable structures is meaningless, and it would not be surprising if these predicted structures were not physically possible. The authors should only discuss predicted ligand structures with pLDDT > 90, and they need to provide examples of physically infeasible structures that AF3 predicts with high reliability.

2. It is already known that in structure prediction using coevolutionary information obtained from multiple sequence alignments, such as AF, the coevolutionary information does not change significantly when amino acid substitutions are introduced. Therefore, AF does not always correctly evaluate the effect of amino acid substitutions and tends to predict stable structures similar to the wild type even for mutants destabilized by amino acid substitutions. This manuscript is a confirmatory study that reiterates this point and lacks novelty.

3. The present study assumes that introducing amino acid substitutions or ligand modifications weakens the protein-ligand interaction. Is there any such experimental evidence for the proteins used here? The authors should use proteins for which there is experimental evidence of different binding affinities for similar but different ligands.

4. When amino acid substitutions are introduced into a protein, authors should use AF3 to predict not only the protein-ligand complex structure, but also the protein structure alone without the ligand to ensure that the structure is not destabilized. If the protein structure is significantly altered by the introduction of amino acid substitutions alone, then it is not possible to discuss ligand binding.

5. When discussing the prediction of the protein-ligand complex structure, the authors only mention the ligand structure in the manuscript. They should also mention whether the protein structure was correctly predicted.

6. In this study, only ATP binding to CDK2 and glucose binding to glucose dehydrogenase were used for analysis. However, there are many other ATP-binding proteins in addition to CDK2. Why did the authors choose this protein?
7. Why did the authors use glucose dehydrogenase instead of CDK2 for the ligand modification analysis? Both amino acid substitutions and ligand modifications should be performed for both proteins to show that similar results can be reproduced.
8. It is too simplistic to try to draw general conclusions based only on calculations for a limited number of proteins. In addition to presenting and discussing detailed data for the three proteins described in the Supplementary Information, similar calculations should be performed for many more proteins (e.g., hundreds more) and for different types of ligands (small molecules, nucleic acids, metals, etc.) to confirm their reproducibility.
9. Line 153: The results in Figures S2 and S3 should also be discussed in the text.
10. Line 95: Please describe how many structures were predicted for each. It is also necessary to show that the sampling was sufficient for this analysis. Multiple structure predictions should be made and the distribution of ligand positions should be discussed.
11. Authors should verify that the results of the AF3 predictions are reproduced when multiple predictions are made using a seed other than 1.
12. Line 135: RFAA and Boltz-1 do not even correctly predict the ligand structure before modification. In such cases, analysis of ligand modifications would be meaningless.
13. It would be better to quantitatively evaluate the similarity before and after ligand modification using the Tanimoto coefficient or other methods.
14. Instead of using the standard method of AF3 prediction, could we develop a method of using AF3 that would allow us to more accurately predict the complex structure when the ligand structure is slightly altered?
15. Can we use the AF3 output structure as a starting structure and then perform physics-based calculations to correctly evaluate the effects of amino acid substitutions and ligand modifications?
16. Line 140: Where in the protein did the ligand bind that was out of the original pocket?
17. This reviewer is curious about the results obtained when the same analysis as in this study is performed on PDB structures not included in the training data of AF3. It is very possible that the predicted structures of modified ligands are the same as those obtained experimentally. If so, is this called overfitting? Can you show such examples?
18. In the funnel metadynamics simulation, the time evolution of the RMSD from the initial ligand position should be shown.
19. It appears that the 1000 ns funnel metadynamics simulation was performed only once for each structure. However, the same calculation should be repeated at least three times to check the reproducibility.
20. The calculation method of the funnel metadynamics simulation needs to be described in more detail.
21. What does "Wild-Type" in Table 1 refer to, the PDB structure or the wild-type structure predicted by AF3?
22. Supplementary Information Figures should be numbered in the order in which they appear in the text.
23. "Mutating to dissimilar residues" and "Funnel metadynamics" in the Supplementary Methods should be moved to the Methods section of the main text.
24. Figure 1: The figure legend should indicate which color represents which structure.
25. Figure 3A: A color bar should be shown to indicate the relationship between the surface charge and the color shown.
26. Figure 3 caption: using the using the  using the
27. In Figure 3, what do the white stick structures shown in all panels indicate?
28. Page 1 of Supplementary Information, section 1.2, line 4: $\delta_{pij} \rightarrow \delta_{vij}$

Version 1:

Reviewer comments:

Reviewer #1

(Remarks to the Author)

The authors have addressed my concerns adequately.

Reviewer #2

(Remarks to the Author)

The authors have successfully addressed my comments.

Reviewer #3

(Remarks to the Author)

In the revised manuscript, the authors performed additional calculations, particularly analyses of the CASF dataset (285 protein-ligand complexes) and the MEK1 structure that was not used to train co-folding models. These revisions adequately address my previous comments, and the manuscript has improved substantially. However, please address the following minor issues and typos.

1. Figure 4 appears before Figure 3 in the main text.
2. The image quality in Figure 2 is poor.
3. In Figure 5A, superscript "2" in e/A^2
4. Figures S5 and S6: The unit of RMSD (A) should be shown as an angstrom.
5. Figures S13 and S14: Label the horizontal axis as RMSD (A), where A is an angstrom.
6. Figures S7-S12, S15 and S16: Add units (angstroms) to RMSD.
7. Figure 4 caption: "A" in "RMSD < 2 A" should be shown as an angstrom. The same applies to the vertical axis label.
8. Typographical errors
Please carefully check the manuscript for typos, including those listed below.

In the main text:

Line 252: unsurprising  unsurprising

Line 259: oin order to  in order to

Line 376: 1000ns  1000 ns (add a space)

In the Supplementary file:

Line 5 of section 1.1: mutagensis  mutagenesis

Line 4 of section 1.2: strutural  structural

Line 4 of section 2.1: showns  shown

Line 3 from the bottom of section 2.1: predicions  predictions

Line 1 from the bottom of section 2.1: eachother  each other

Line 5 of section 2.3: indicates  indicate

Line 7 of section 2.3: models'  models

Line 5 of section 2.4: demonstrates  demonstrate

Response to Reviewers

We would like to thank the reviewers for their careful reading of the manuscript, as well as their valuable feedback and suggestions for improvement. We have addressed the comments point by point as detailed below, and we hope that our revisions meet the reviewers' expectations.

Reviewer #1

Since AF3 and related methods were designed to predict structure for protein-ligand pairs that are known to interact, their inability to distinguish true interactors from artificially designed non-interactors (as tested here) is perhaps unsurprising. As the authors note, a key limitation is the scarcity of training data, a well-recognized challenge in the field. However, for practical applications, the confidence metrics provided by these methods are crucial and should be considered in the benchmark assessment. Low-confidence predictions could alert users to unreliable results, preventing misinterpretation. While the authors include these metrics in supplementary tables, they do not mention nor discuss them at all. A deeper analysis of confidence scores versus the degree of adverse modification would strengthen the study.

We appreciate this suggestion and have incorporated a detailed analysis of confidence scores and prediction accuracy under varying degrees of adverse modification in the CASF dataset, which comprises 285 diverse protein–ligand complexes. Our results show that, irrespective of the type of modification, more than 50% of systems correctly predicted by AlphaFold3 (ligand RMSD < 2 Å) retain the same predicted ligand pose even after binding site disruption. As now discussed in the manuscript, this proportion of conserved binding modes is largely independent of the confidence score. For example, among complexes with a confidence score >80, between 42% and 52% maintain an unaltered ligand pose depending on the disruption applied (compared to 78% unaltered in the absence of disruption).

We also clustered the systems by protein family and examined correlations between pose RMSD and various physicochemical properties of the binding site and bound ligand, but did not observe any consistent trend indicating when AlphaFold3 is more likely to preserve the same pose. The results of this analysis are presented in the Supplementary Information Figures S13, S14, and S15.

A complicating factor to this study is the biological robustness of protein-ligand systems: some modifications may be physically and functionally tolerated, while others may not. The authors use funnel-metadynamics simulations to assess bound states of their modified ligands, and some of the modifications still permit (or even favor) bound states according to their simulations. In such cases, are the AF3 predictions physically plausible? This distinction warrants further discussion.

We have expanded the discussion on the results of the funnel metadynamics simulations. In particular, we address the plausibility of AlphaFold3 predictions in cases where a bound state is still accessible despite binding site disruptions, as revealed by the funnel metadynamics.

In the Discussion, the authors propose that Rosetta energy scores or MD free energy calculations could provide "more rigorous confirmation." While cross-validation with physics-based methods may be useful, these approaches have well-known limitations. Otherwise, deep learning would not lead the field now. A more pertinent question is whether these methods can complement each other, which this study does not explore sufficiently. While the adversarial cases here can be dismissed due to obvious clashes, what about physically plausible predictions without any clashes for novel ligands? Are they hallucinations or realistic predictions?

Due to the high computational cost, the current manuscript rigorously evaluates the physical plausibility of co-folding predictions using funnel metadynamics for only a few selected cases. While simpler physical assessments based on force field energies or physics-based scoring functions are computationally more feasible, they, as the reviewer rightly noted, have limitations in accuracy and cannot provide a fully reliable judgment. To provide further supporting evidence to this statement, we ran computationally efficient scoring schemes on the same systems for which funnel metadynamics was performed: Single-point MM/GBSA calculations and blind re-docking. MM/GBSA calculations were done using OpenMM by minimizing the predicted complex and evaluating its potential energy, then separating the protein and ligand components, minimizing, and evaluating their potential energies. The energies of the individual proteins and ligand energies are subtracted from the complex energy to get an estimated dG. The blind re-docking was performed using Smina with a bounding box encompassing the full protein on the predicted structures with ligand removed.

System	MM/GBSA dG (kJ/mol)	Smina Redocking (RMSD)
Binding Site Wild-Type	-193.5	2.7
Binding Site Removal	-198.1	2.6
Binding Site Packing	-236.2	2.3
Binding Site Inversion	-34.4	9.4
Glucose Wild-Type	-158.5	13.8
Glucose 1x Methyl	-166.3	4.5
Glucose 2x Methyl	-170.3	4.3
Glucose 3x Methyl	156.3	1.1
Glucose 4x Methyl	-122.0	12.5
Glucose 5x Methyl	-98.8	10.5
ATP 1x tert-butyl	-52.6	8.2

ATP 2x tert-butyl	-289.5	2.2
ATP 3x tert-butyl	192.1	2.9
ATP 1x quaternary amine	-128.4	3.3
ATP 2x quaternary amine	-25.2	2.4
ATP 3x quaternary amine	-244.9	1.9

As can be seen in the above results, there is no correlation observed between funnel metadynamics and the MM/GBSA or redocking results; for example, there are some modified ligands (glucose with one or two methyls, or ATP with three quaternary amines) that shows improved binding compared to the wild type when we would expect to see a large, positive $\Delta\Delta G$ difference. The Smina results also indicate that docking is not robust enough to distinguish modified and unmodified systems; for example, there are several cases where Smina retrieves a near-native pose despite the system being heavily mutated (glucose with three methyls, ATP with three quaternary amines).

Therefore, more rigorous free energy methods, such as absolute FEP or funnel metadynamics, would be necessary for a comprehensive evaluation. However, such simulations are computationally intensive and would constitute a separate future project beyond our current resource capabilities.

Nevertheless, there is additional evidence suggesting that co-folding models may underperform in predicting complexes involving previously unseen proteins or ligands. In such scenarios, traditional physics-based methods—including simple docking tools like AutoDock Vina—have been shown to outperform co-folding approaches. We have added a discussion of this point along with a citation to the relevant study (<https://www.biorxiv.org/content/10.1101/2025.02.03.636309v1.full.pdf>) in the revised manuscript.

Minor Comments:

1. Please include at least one reference for funnel-metadynamics.

We have now added additional details and references relating to the funnel metadynamics simulations.

2. In Figures 1–3, consider annotating unphysical interactions/clashes for clarity.

For sake of clarity, we have added some additional details to the text remarking on unphysical interactions/cases seen within the figure.

3. In Figure S1, the labels are too small to read. Please also expand the caption to explain the color scheme and mark any unphysical interactions, if present.

As requested, we have expanded the caption of these figures to explain the color scheme and describe any unphysical interactions. The label size could not be increased without separating the figure into multiple figures spanning multiple pages. Fortunately, the image quality is high and labels can be read by zooming into the image.

Reviewer #2 (Remarks to the Author):

The first, major point is that the article is at best a compilation of two case studies. While I find the authors' methodology and results very compelling, I would be hard-pressed to present this article to a colleague to support the conclusions -- there simply isn't enough evidence. The conclusions of the article would be far stronger if they repeated some of their experiments on multiple proteins and ligands. Even if they repeated the binding site removed/packed experiments on all crystal structures on CASF16 (~200 structures), that would be enough to provide a broad view across target classes and compounds (though I would probably suggest that the authors also explore more ligand modifications). This experiment would be trivial to implement and require at most a couple of GPU days with even a gaming-grade GPU, so I cannot see any reason to not perform it. In my opinion, as much as I think this is a promising article with a valuable message for the molecular biology community, without a broader set of experiments I could not in good faith recommend publication on Nature Communications, or other peer-reviewed journals.

Thank you for this recommendation. We have taken your advice to expand our analysis to the CASF16 dataset of protein-ligand structures. We have automated the binding site mutation process and performed structural predictions with AlphaFold3 across the full dataset. We have added these results to the manuscript and discussed their impact. Overall, we observed the same pattern of unphysical behavior across the majority of systems in the dataset. In detail, we observed that regardless of the type of modification, over 50% of systems correctly predicted by AlphaFold3 (with ligand RMSD < 2 Å) retain the same predicted ligand pose even after binding site disruption. This proportion of conserved binding modes is largely independent of the confidence score. For example, among complexes with a confidence score >80, between 42% and 52% retained an unaltered ligand pose depending on the type of disruption applied (compared to 78% remaining unaltered when no disruption was introduced).

We also clustered the systems by protein family and analyzed correlations between pose RMSD and various physicochemical properties of the binding site and bound ligand. However, we did not observe any consistent trends that would indicate under which conditions AlphaFold3 is more likely to preserve the same pose.

The second and third point are more minor. First, the authors have not fully disclosed the methodology: there is no mention to how the funnel metadynamics simulations were conducted, which is important to the conclusions of the article.

We have now added additional details and discussion concerning the funnel metadynamics simulations to Section 4.3 of the methodology.

Second, the authors have forgotten to cite some important literature in the area of studying the physical underpinnings of AI structure prediction (some of which include my own work -- sorry, I hate to be that guy):

Outeiral, C., Nissley, D.A. and Deane, C.M., 2022. Current structure predictors are not learning the physics of protein folding. *Bioinformatics*, 38(7), pp.1881-1887.

Chakravarty, D., Schafer, J.W., Chen, E.A., Thole, J.F., Ronish, L.A., Lee, M. and Porter, L.L., 2024. AlphaFold predictions of fold-switched conformations are driven by structure memorization. *Nature communications*, 15(1), p.7296.

Thank you very much for these suggestions. These citations have now been properly included in the manuscript.

Reviewer #3 (Remarks to the Author):

1. In the AF prediction, the main chain structure is expected to be correct when pLDDT > 70, and both the main chain and side chain structures are expected to be correct when pLDDT > 90. Supplementary Tables S2-S4 show that in the present predictions, pLDDT was greater than 90 for the wild type. In contrast, pLDDT was less than 90 for the structures predicted when amino acid substitutions or ligand modifications are introduced, suggesting that side-chain and ligand structures are unreliable for these predictions. The authors made the argument that "physics does not hold" with unreliable ligand structures. However, the argument about unreliable structures is meaningless, and it would not be surprising if these predicted structures were not physically possible. The authors should only discuss predicted ligand structures with pLDDT > 90, and they need to provide examples of physically infeasible structures that AF3 predicts with high reliability.

We recognize that pLDDT values provide a valuable metric for the model's confidence for a given structural prediction, however we would resist the suggestion that we should only discuss predicted ligand structures with pLDDT > 90 for several reasons:

1. It is rare to get a predicted structure where the mean ligand pLDDT value is >90, even for wild type structures. For example, in our evaluation of the 285 CASF structures, which were included in the training, are predicted correctly by the model (~80% within 2Å RMSD to native pose), yet less than 5% of the structures had a mean ligand pLDDT >90.
2. The authors of AlphaFold3 and other co-folding models do not make this distinction. When presenting results, they do not only discuss the results that had very high pLDDT values, they discuss all results regardless of pLDDT.
3. Ligand pLDDT is a relatively new metric with the introduction of co-folding models that is not yet fully understood. As you remarked, in protein predictions it's generally accepted that pLDDT >70 indicates that the backbone is properly placed, but side-chains may not

be. It is unclear at this point exactly how this translates to ligand predictions, where tokens represent individual atoms rather than entire residues.

4. Users of these tools will undoubtedly use predicted structures that have pLDDT values less than 90. It is valuable to understand and discuss how these structural predictions are made and what their limitations are for subsequent research.

In our analysis on the CASF dataset, we made distinct statistics for different ranges of pLDDT values but observed similar behavior upon binding site disruption independent of pLDDT value.

2. It is already known that in structure prediction using coevolutionary information obtained from multiple sequence alignments, such as AF, the coevolutionary information does not change significantly when amino acid substitutions are introduced. Therefore, AF does not always correctly evaluate the effect of amino acid substitutions and tends to predict stable structures similar to the wild type even for mutants destabilized by amino acid substitutions. This manuscript is a confirmatory study that reiterates this point and lacks novelty.

We agree that several studies have investigated the lack of sensitivity of co-folding methods on amino acid substitutions for predicting protein structures. However, we disagree that the current study reiterates this points for the following reasons:

1. Co-evolutionary information for protein-ligand interactions does not exist and therefore should not guide the placement of a small-molecule ligand in the binding site of a protein.
2. Much work has been done to analyze the conventional protein folding models such as AlphaFold2. However, in this work, we specifically investigated if co-folding models understand the physics of protein-ligand interactions as this is an important, yet poorly understood aspect of these models. This includes introducing additional mutations on the ligand side that were not possible with previous models.
3. There are significant differences between the co-folding models in their use of co-evolutionary information. For example, Chai is not making use of co-evolutionary information seen in multiple sequence alignments, yet suffers from the same lack of physical robustness. Therefore, our study indicates problems beyond simply the use of MSA data.

3. The present study assumes that introducing amino acid substitutions or ligand modifications weakens the protein-ligand interaction. Is there any such experimental evidence for the proteins used here? The authors should use proteins for which there is experimental evidence of different binding affinities for similar but different ligands.

We of course do not have any experimental data to confirm the non-binding nature of the modifications we make. Although extremely unlikely, there is always a small possibility that, despite the drastic changes in the binding sites, the structures predicted by these tools are still the lowest energy conformation. Our funnel metadynamics simulations on a subset of systems,

however, showed that binding is no longer energetically feasible under those drastic mutation schemes.

4. When amino acid substitutions are introduced into a protein, authors should use AF3 to predict not only the protein-ligand complex structure, but also the protein structure alone without the ligand to ensure that the structure is not destabilized. If the protein structure is significantly altered by the introduction of amino acid substitutions alone, then it is not possible to discuss ligand binding.

This is an excellent suggestion. We predicted the protein structure without ligand for the different substitutions. Figure S16 in the revised Supplementary Information demonstrates that AF3 still predicts in 95% of CASF systems the wildtype structure within an RMSD of 1.5 Angstrom.

5. When discussing the prediction of the protein-ligand complex structure, the authors only mention the ligand structure in the manuscript. They should also mention whether the protein structure was correctly predicted.

We have added some additional clarifying details to the manuscript. With the exception of binding site sidechains, the tertiary protein structure was predicted with high accuracy independent of the substitution pattern without exception.

6. In this study, only ATP binding to CDK2 and glucose binding to glucose dehydrogenase were used for analysis. However, there are many other ATP-binding proteins in addition to CDK2. Why did the authors choose this protein?

There was no particular reason for the choice of CDK2 over other ATP-binding proteins. Also to address other reviewers' comments, we have extended our analysis to the CASF16 dataset of 285 protein-ligand complexes (including several other kinases), most of which demonstrate the same behavior.

7. Why did the authors use glucose dehydrogenase instead of CDK2 for the ligand modification analysis? Both amino acid substitutions and ligand modifications should be performed for both proteins to show that similar results can be reproduced.

The choice to go with another system was intended to demonstrate that this behavior is not limited to such specific systems and can be replicated regardless of a specific system. Ligand modifications were done on ATP-ligand binding to CDK2 and can be viewed in Figure 5.

8. It is too simplistic to try to draw general conclusions based only on calculations for a limited number of proteins. In addition to presenting and discussing detailed data for the three proteins described in the Supplementary Information, similar calculations should

be performed for many more proteins (e.g., hundreds more) and for different types of ligands (small molecules, nucleic acids, metals, etc.) to confirm their reproducibility.

Following this suggestion and the previous suggestion by Reviewer #2, we have conducted a systematic experiment of mutating binding sites across a dataset of 285 protein-ligand complexes (CASF16) and present the results in the paper. Overall, the behavior is consistent across the majority of all systems.

9. Line 153: The results in Figures S2 and S3 should also be discussed in the text.

Additional details describing the results in Figures S2 and S3 have been added to the text in Section 2.1 of the Supplementary Information.

10. Line 95: Please describe how many structures were predicted for each. It is also necessary to show that the sampling was sufficient for this analysis. Multiple structure predictions should be made and the distribution of ligand positions should be discussed.

Unless otherwise noted, we used the default setting. For AlphaFold3 and Chai-1 that means five structures are predicted. Out of these, we always use the structure with the highest confidence. For RFAA and Boltz-1, only a single structure is generated by default. These details are described in the SI.

11. Authors should verify that the results of the AF3 predictions are reproduced when multiple predictions are made using a seed other than 1.

On a small set of systems we confirmed that the behavior of AlphaFold3 persists, regardless of the input seed.

12. Line 135: RFAA and Boltz-1 do not even correctly predict the ligand structure before modification. In such cases, analysis of ligand modifications would be meaningless.

It is not that RFAA and Boltz-1 fail entirely on this system. The ligand is still placed in the binding site and forms many of the same hydrogen bonding interactions as the native structure. However, since the ligand is glucose and behaves rather symmetrically, these programs have rotated the ligand within the binding site such that the RMSD is over 2.0 Å. We still believe this example is useful to include in the manuscript to observe how the different co-folding models behave. For further validation of these wildtype predictions, we visualized the funnel metadynamics MD simulation of the wildtype structure and observed rotation of the glucose in a similar way to the RFAA and Boltz predictions.

13. It would be better to quantitatively evaluate the similarity before and after ligand modification using the Tanimoto coefficient or other methods.

This has been added as Section 2.5 to the Supplementary Information of the article. Tanimoto similarity coefficients of the mutated ligands range from between 0.143 to 0.585.

14. Instead of using the standard method of AF3 prediction, could we develop a method of using AF3 that would allow us to more accurately predict the complex structure when the ligand structure is slightly altered?

It is an interesting question, how we can make use of the existing AlphaFold model to make more accurate predictions when we know the ligand structure is altered. It probably would require either direct physics-integration (which is not obvious to implement within AF3) or to have extensive SAR datasets with structural data and also potentially including negative data. This is beyond the scope of the current article and we currently have not developed any method of using the existing AF3 model in a more reliable way.

15. Can we use the AF3 output structure as a starting structure and then perform physics-based calculations to correctly evaluate the effects of amino acid substitutions and ligand modifications?

That is our guidance at this point: co-folding models can be useful for generating ideas or proposals. However, physics-based calculations are essential for correctly evaluating the effects of amino acid substitutions and ligand modifications. While developing a rigorous methodology for the evaluation of amino acid substitutions and ligand modifications is beyond the scope of this current work, we believe it will be essential for future developments.

16. Line 140: Where in the protein did the ligand bind that was out of the original pocket?

The ligand was binding just outside of the pictured pocket, still near but not placed within the binding pocket. All predicted structures will be made available as supplementary data on Zenodo which can be accessed here: <https://zenodo.org/records/14749304>

17. This reviewer is curious about the results obtained when the same analysis as in this study is performed on PDB structures not included in the training data of AF3. It is very possible that the predicted structures of modified ligands are the same as those obtained experimentally. If so, is this called overfitting? Can you show such examples?

This is a great suggestion. We have now extended our analysis to specifically include an example not seen during training (PDB: 7XLP) now presented in Figure 2 of the main article (with associated discussion and Figure S2 and Table S2 in the SI). The results agree with the rest of the study, with very little perturbation in the ligand positioning after modifying all binding site residues, with the exception of RFAA, which was unable to predict any structures correctly (including the wildtype). This system was identified via the Runs N' Poses dataset which calculated a similarity value (sucos_shape_pocket_qcov) of 46.21, indicating low similarity to the training set. Since the behavior is still seen in novel structures not present during training, we have moved away from using the term "overfitting" as much in the article.

18. In the funnel metadynamics simulation, the time evolution of the RMSD from the initial ligand position should be shown.

We have now included the time evolution of the RMSD during the funnel metadynamics in the Supplementary Information Figures S5 and S6.

19. It appears that the 1000 ns funnel metadynamics simulation was performed only once for each structure. However, the same calculation should be repeated at least three times to check the reproducibility.

We have repeated the funnel metadynamics simulations reported in the article to include replicates of each simulation. The results are in agreement with the originally presented results. We have added the results for the simulation replicates to the manuscript.

20. The calculation method of the funnel metadynamics simulation needs to be described in more detail.

Additional details describing the funnel metadynamics simulations have been added to the manuscript (Section 4.3).

21. What does "Wild-Type" in Table 1 refer to, the PDB structure or the wild-type structure predicted by AF3?

It refers to the wild-type structure predicted by AlphaFold3. An additional comment was added to the manuscript to clarify this fact.

22. Supplementary Information Figures should be numbered in the order in which they appear in the text.

This has been corrected in the manuscript.

23. "Mutating to dissimilar residues" and "Funnel metadynamics" in the Supplementary Methods should be moved to the Methods section of the main text.

This has been corrected in the manuscript.

24. Figure 1: The figure legend should indicate which color represents which structure.

This has been added to the figure for better interpretability.

25. Figure 3A: A color bar should be shown to indicate the relationship between the surface charge and the color shown.

A colorbar has been added to the figure for better interpretability.

26. Figure 3 caption: using the using the  using the

This has been corrected in the manuscript.

27. In Figure 3, what do the white stick structures shown in all panels indicate?

The white sticks represent the position of the unmodified ATP molecule cocrystalized in the original crystal structure. The caption has been updated to reflect this.

28. Page 1 of Supplementary Information, section 1.2, line 4: δ_{pij}  δ_{vij}

This has been corrected in the manuscript. It is now part of the main manuscript (Section 4.1).

NCOMMS-25-17262

Response to Reviewer 3

1. Figure 4 appears before Figure 3 in the main text.

The numbering of Figure 3 and 4 has now been swapped with the text updated accordingly.

2. The image quality in Figure 2 is poor.

The original image was now replaced with one at a much higher resolution.

3. In Figure 5A, superscript "2" in e/A^2

This typography issue has now been corrected.

4. Figures S5 and S6: The unit of RMSD (A) should be shown as an angstrom.

This issue has been corrected and units are shown in angstrom.

5. Figures S13 and S14: Label the horizontal axis as RMSD (A), where A is an angstrom.

This issue has been corrected and properly labeled axes have been added to the figures.

6. Figures S7-S12, S15 and S16: Add units (angstroms) to RMSD.

The units (angstroms) have been added to each of the figures.

7. Figure 4 caption: "A" in "RMSD < 2 A" should be shown as an angstrom. The same applies to the vertical axis label.

This has been corrected in both the caption and figure image.

8. Typographical errors

Please carefully check the manuscript for typos, including those listed below.

In the main text:

Line 252: unsuprising  unsurprising

Line 259: oin order to  in order to

Line 376: 1000ns  1000 ns (add a space)

In the Supplementary file:

Line 5 of section 1.1: mutagensis  mutagenesis

Line 4 of section 1.2: strutural  structural

Line 4 of section 2.1: showns  shown

Line 3 from the bottom of section 2.1: predicions  predictions

Line 1 from the bottom of section 2.1: eachother  each other

Line 5 of section 2.3: indicates  indicate

Line 7 of section 2.3: models'  models

Line 5 of section 2.4: demonstrates  demonstrate

All of the above typographical errors have been corrected in the main text and supplementary file.